# Disentangling the mutational effects on protein stability and interaction of human MLH1

Sven Larsen-Ledet©*, Aleksandra Panfilova, Amelie Stein©*

Department of Biology, University of Copenhagen, Copenhagen, Denmark

* sven.larsenledet@bio.ku.dk (SL-L); amelie.stein@bio.ku.dk (AS)

## Abstract

Missense mutations can have diverse effects on proteins, depending on their location within the protein and the specific amino acid substitution. Mutations in the DNA mismatch repair gene *MLH1* are associated with Lynch syndrome, yet the underlying mechanism of most disease-causing mutations remains elusive. To address this gap, we aim to disentangle the mutational effects on two essential properties for MLH1 function: protein stability and protein-protein interaction. We systematically examine the cellular abundance and interaction with PMS2 of 4839 (94%) MLH1 variants in the C-terminal domain. Our combined data shows that most MLH1 variants lose interaction with PMS2 due to reduced cellular abundance. However, substitutions to charged residues in the canonical interface lead to reduced interaction with PMS2. Unexpectedly, we also identify a distal region in the C-terminal domain of MLH1 where substitutions cause both decreased and increased binding with PMS2, and propose a region in PMS2 as the binding site. Our data correlate with clinical classifications of benign and pathogenic MLH1 variants and align with thermodynamic stability predictions and evolutionary conservation. This work provides mechanistic insights into variant consequences and may help interpret MLH1 variants.

## Author summary

Mutations in proteins, including single amino acid substitutions, can disrupt normal cellular function and lead to diseases. Hence, there is great interest in understanding their consequences from both basic science and clinical genetics perspectives. Loss-of-function variants in the human mismatch repair protein MLH1 are associated with Lynch syndrome, a hereditary condition that significantly increases the risk of certain cancers. Detailed insights into the molecular consequences of MLH1 variants are essential for understanding the mechanisms underlying Lynch syndrome. In this study, we use high-throughput experimental approaches to test how missense mutations affect the cellular abundance of MLH1 and the interaction with its partner, PMS2. We find that many variants

**Data availability statement:** All data generated are included in the figures and supplemental files. Sequencing reads are available at the NCBI Gene Expression Omnibus (GEO) repository (accession number: GSE273652). Processed cellular abundance and PMS2 interaction scores are available in MaveDB (accession number: urn:mavedb:00001218-a). Data and code are available at GitHub (https://github.com/KULL-Centre/_2024_Larsen-Ledet_MLH1).

**Funding:** The present work was supported by grants from the Lundbeck Foundation (R272-2017–452, to A.S.) and the Novo Nordisk Foundation challenge program PRISM (NNF18OC0033950, to A.S.). S.L-L. and A.P. received salary from R272-2017–452. The funders had no role in study design, data collection and analysis, decision to publish, or preparation of the manuscript.

**Competing interests:** The authors have declared that no competing interests exist.

disrupting the interaction with PMS2 also lead to reduced cellular abundance levels, suggesting a strong relationship between protein stability and interaction in MLH1. Additionally, we identify a potential novel interaction site between MLH1 and PMS2, where mutations in MLH1 either increase or decrease interaction without affecting abundance. Our datasets are in line with the classification of clinically annotated MLH1 variants and may aid in interpreting variants of uncertain significance, while also providing new insights into the molecular details of the interaction with PMS2.

## Introduction

Proteins constitute the functional units of the cell and perform essential cellular processes to sustain cell function and health. To obtain biological function, most proteins must fold into a marginally stable native structure, which is largely determined by the amino acid sequence [1]. This makes proteins extremely sensitive to missense mutations that disrupt the structural stability, leading to protein degradation and reduced cellular abundance. Thus, protein stability is fundamental for protein function and, indeed, the most frequent mechanism of disease-causing missense mutations is through destabilization and degradation [2–9]. However, in addition to protein stability, proteins often have other properties important for their function. The function of most proteins relies on interactions with other proteins to form complexes and networks, making protein-protein interactions (PPIs) a central property of protein function [10–12].

In general, PPIs can be grouped into obligate or non-obligate interactions based on whether the proteins function only as a complex or can also function as monomers [13,14]. Non-obligate interactions are often of lower affinity, transient, and involved in regulatory and signaling pathways, whereas obligate PPIs are typically stronger with larger and more hydrophobic interfaces [14–16]. Several studies have shown that disease-causing missense mutations are enriched in PPI interfaces [17–20], and it has been estimated that the underlying mechanism of around half of all disease-causing missense mutations in proteins that interact with other proteins is through the perturbation of specific interfaces [21–23]. This suggests that numerous clinically relevant protein variants affecting PPI interfaces remain stable in the cell, which highlights that mutations may affect protein stability and PPIs differently, depending on the type and site of the mutation. Mechanistic insights into mutational effects on protein stability and interactions have important implications for our understanding of protein function and diseases, and may provide avenues towards developing treatments.

MutL homolog 1 (MLH1) is a 756-residue protein that folds into an N- and a C-terminal domain, separated by a long intrinsically disordered linker [24]. MLH1 forms an obligate heterodimer with postmeiotic segregation increased 2 (PMS2), termed MutLα. Although the structure of the human MutLα complex remains unresolved, it is known that MLH1 and PMS2 dimerize through their C-terminal domains

[25]. MutLα participates in the conserved DNA mismatch repair (MMR) pathway that detects and repairs base-base mismatches during DNA replication [26,27]. Defects in the MMR pathway increase the risk of spontaneous somatic mutations and sporadic cancers. Loss of function mutations in MLH1 cause Lynch syndrome, an inherited disease that predisposes individuals to various cancer types, such as colorectal and endometrial cancer [28,29]. A large proportion of Lynch syndrome patients suffer from pathogenic missense mutations in the *MLH1* gene [30,31]. However, the mechanistic effects of most of these variants are still unknown, and most of the MLH1 variants observed in the population are still categorized as variants of uncertain significance (VUS) [32–34].

Here, we use orthogonal yeast-based growth assays to disentangle the mutational effects on the abundance of MLH1 and on its interaction with PMS2. We systematically screen site-saturated libraries comprising 4839 (94%) missense and nonsense variants in the C-terminal domain of human MLH1. We find that MLH1 variants predominantly affect protein abundance compared to interaction and that these positions often are confined to buried residues. However, substitutions to charged residues in positions in the suggested interface between MLH1 and PMS2 are detrimental to interaction. Surprisingly, we find that a region distal to the interface confers improved interaction without affecting abundance. We find that the experimental results correlate with computational predictions and can distinguish clinically classified pathogenic variants from benign population variants. Our data suggest that single missense mutations are more likely to affect abundance than interaction but do pinpoint specific positions critical for interaction in known and novel interfaces with PMS2.

## Results

### Yeast-based assays to measure MLH1 abundance and interaction with PMS2

To disentangle the mutational effects on protein abundance and interaction, we selected the interaction between the human proteins MLH1 and PMS2. MLH1 and PMS2 form the MutLα complex through their C-terminal domains, an essential component of the DNA mismatch repair pathway (Fig 1A). The structure of the human C-terminal MutLα complex is unknown, and the C-terminal domain of PMS2 has not been structurally resolved, making it challenging to accurately assign the interface between the two proteins. However, despite the moderate sequence identity of the human C-terminal domains (30% for MLH1 and 41% for PMS2) with their yeast orthologs, MLH1 and PMS1, an AlphaFold2-Multimer-predicted model of the human C-terminal complex exhibits relatively high structural similarity (RMSD = 2.08 Å) to the yeast complex (PDB: 4E4W) [35]. Therefore, we used AlphaFold2-Multimer to model the human C-terminal MutLα complex (Figs 1A, S1). The structure predicted that MLH1 and PMS2 dimerize through four β-strands, three of which are close in sequence, while the fourth is located towards the C-terminus.

To measure the abundance of MLH1 variants, we used a previously described dihydrofolate reductase protein-fragment complementation assay (DHFR-PCA) (Fig 1B) [36–39]. This assay is based on a methotrexate-resistant variant of murine DHFR, which is split into two fragments, F[1,2] and F[3], that are expressed separately but from the same centromere-based vector in wild-type *S. cerevisiae* cells (S2 Fig). The F[3] is fused to MLH1, while the F[1,2] remains free (untagged). Plating of cells on medium with methotrexate (MTX) inhibits the essential endogenous yeast Dfr1 and makes growth dependent on reconstitution of the methotrexate-resistant variant of murine DHFR. A stable, high-abundant MLH1 variant enables reconstitution, allowing growth in the presence of MTX, whereas an unstable, low-abundance MLH1 variant prevents reconstitution, resulting in reduced growth in the presence of MTX. Hence, the DHFR-PCA links cellular abundance of MLH1 to yeast growth. However, this assay solely measures the abundance of MLH1 variants and hence is not amenable to report on other properties important to MLH1 function, such as its interaction with PMS2. Therefore, we used the yeast two-hybrid (Y2H) assay to assess the binding of MLH1 variants to PMS2 (Fig 1C) [40]. In this assay, MLH1 and PMS2 are fused to the activation domain (AD) and DNA binding domain (DBD) of the GAL4 transcription factor, respectively. The two fusion proteins are expressed from their own centromere-based vectors in the MaV203 yeast strain (S3A, S3B Fig). If MLH1 and PMS2 interact, the AD and DBD reconstitute the Gal4 transcription factor and initiate the expression of the *HIS3* reporter gene, enabling growth on medium without histidine.

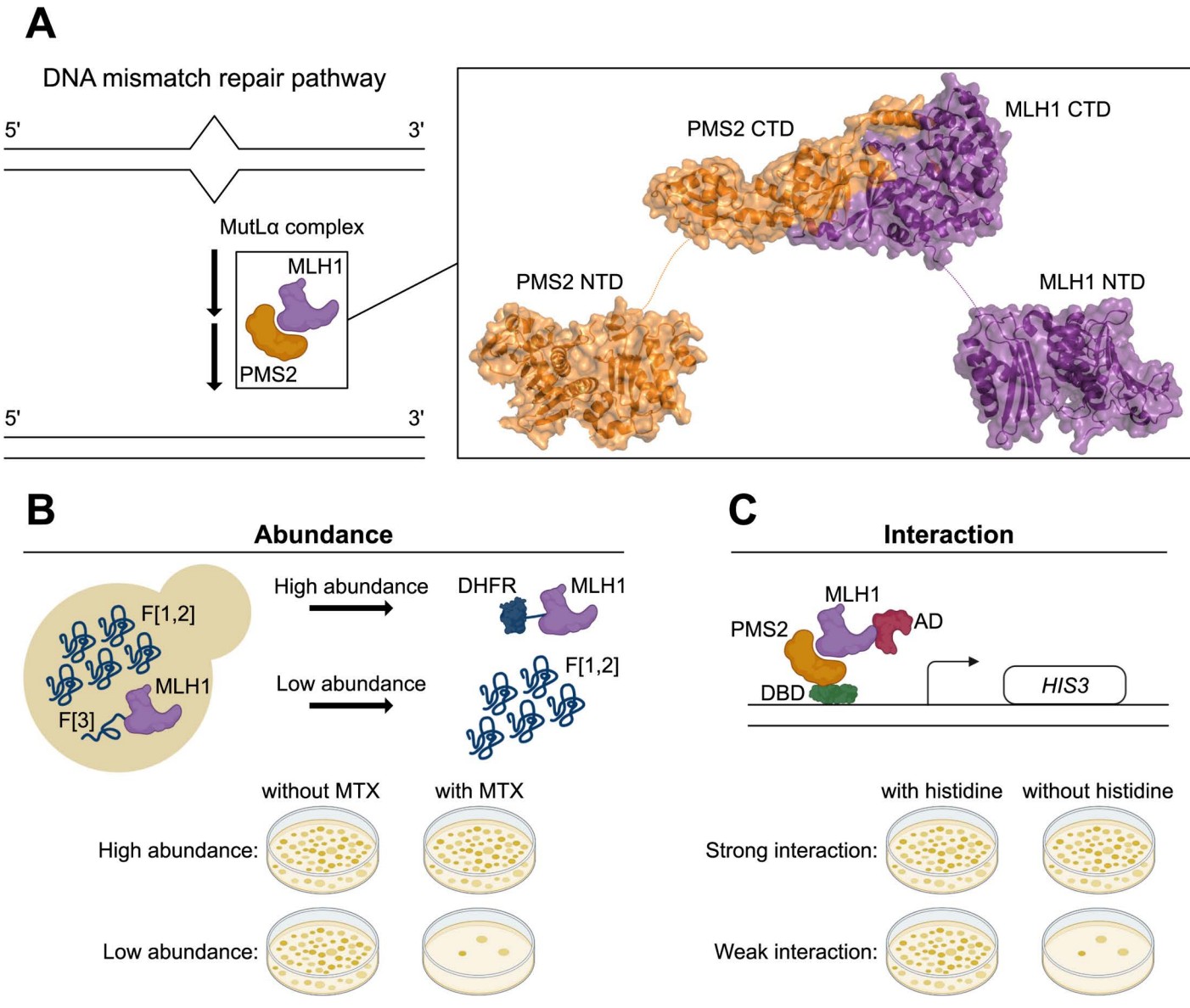

**Fig 1.  _Yeast-based abundance and interaction assays to measure variant effects in MLH1._** (A) The MutLα complex (MLH1:PMS2) participates in the DNA mismatch repair (MMR) pathway together with several other components to recognize and correct mismatches during DNA replication. MLH1 (purple) and PMS2 (orange) fold into an N-terminal domain (NTD) and a C-terminal domain (CTD), separated by a disordered linker. The two proteins interact through their C-terminal domains. The C-terminal MutLα complex was predicted using AlphaFold2-Multimer, whereas the N-terminal domain of MLH1 is PDB: 4P7A and the N-terminal domain of PMS2 is PDB: 1H7S. The disordered linkers between the domains were hand-drawn for illustrative purposes. (B) Schematic illustration of the dihydrofolate reductase protein-fragment complementation assay (DHFR-PCA) used to measure the cellular abundance of MLH1 variants. (C) Schematic illustration of the yeast two-hybrid (Y2H) assay used to measure the interaction of MLH1 variants with PMS2. Panels B and C of Fig 1 and panel B of Fig 2 were created using BioRender.com.

In order to examine the sensitivity and dynamic range of the DHFR-PCA and Y2H assay, we performed small-scale yeast growth assays of the wild-type and five MLH1 variants (Fig 2A). The Q542L and L550P variants locate to the C-terminal domain of MLH1, the S406N variant locates to the intrinsically disordered linker and the I219V and T117M variants locate to the N-terminal domain of MLH1. The S406N and I219V variants are classified as benign in the ClinVar

PLOS Genetics

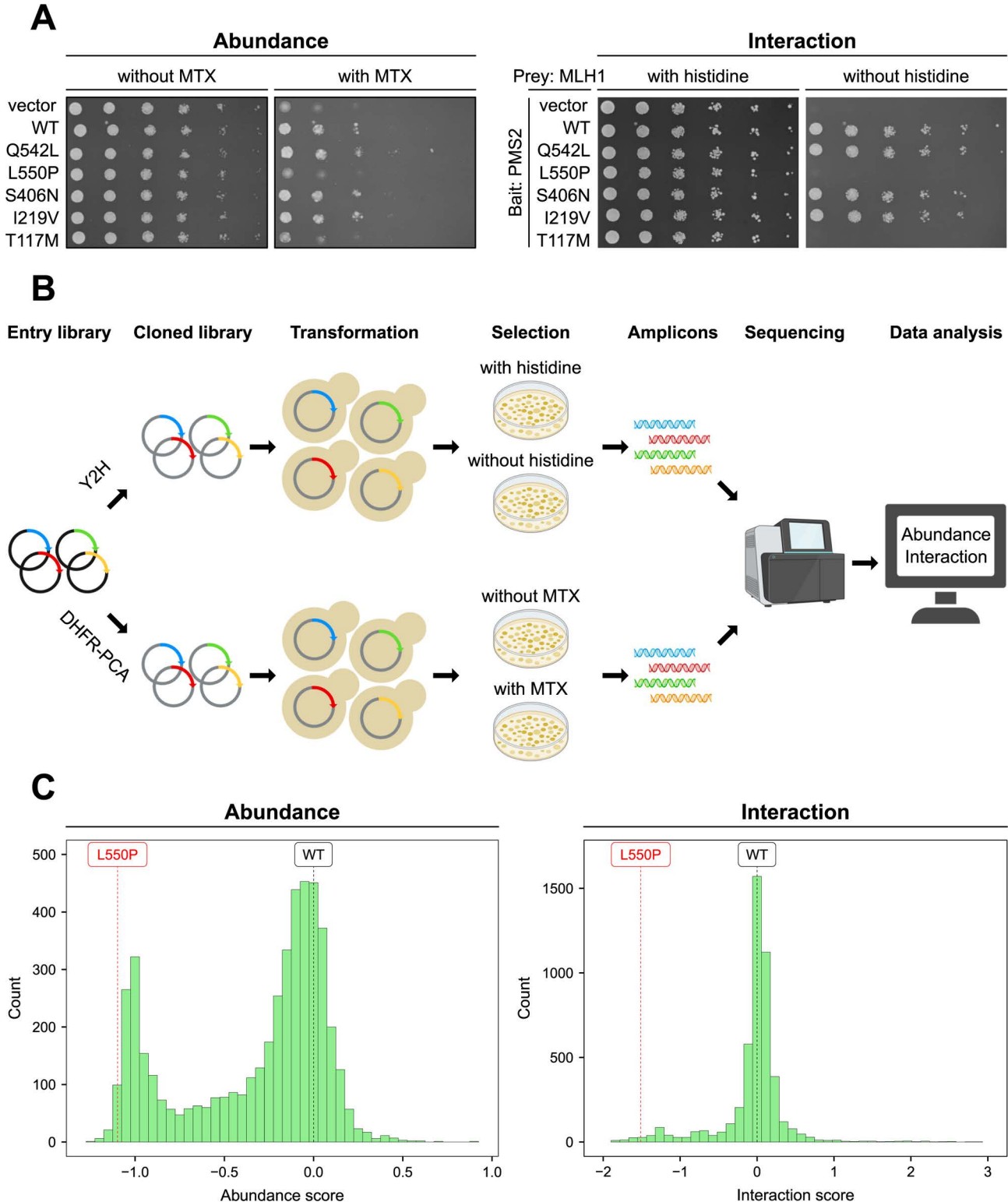

**Fig 2. Screening of site-saturation mutagenesis libraries in the C-terminal domain of MLH1.** (A) Small-scale yeast growth assays to test the DHFR-PCA (left) and the Y2H assay (right). Yeast cells expressed either a vector, wild-type MLH1, or the indicated MLH1 variant. For the DHFR-PCA the vector was an empty vector, while for the Y2H assay the vector was the activation domain (AD) expressed without MLH1. For the DHFR-PCA, cells

were grown on medium with or without MTX at 30 °C. For the Y2H assay, cells co-expressed PMS2 as bait fused to the DNA binding domain and were grown on medium with or without histidine at 30 °C. Q542L and L550P are in the C-terminal domain, S406N is in the disordered linker, and I219V and T117M are in the N-terminal domain. (B) Overview of the experimental design to screen libraries of MLH1 variants for abundance and interaction with PMS2. Parts of this figure were created using BioRender.com. (C) Histograms showing the score distributions of MLH1 variants in the DHFR-PCA (left) and the Y2H assay (right). A score of 0 corresponds to the wild-type in both assays, while a negative or positive score indicates reduced or increased abundance or interaction, respectively. As a reference, the abundance and interaction score of the wild-type MLH1 (black) and the L550P variant (red) are highlighted with dashed lines. The abundance scores were rescaled to range from −1 to 1 (see Materials and Methods).

database [41] and frequently observed in the population, as reported in the Genome Aggregation Database (gnomAD) [42]. The L550P and T117M variants are known to cause Lynch syndrome, are classified as pathogenic, and are extremely rare in the population. The Q542L variant is classified as a VUS and has been shown to exhibit wild-type-like abundance, interaction with PMS2, and MMR activity [25,31,43], although some studies suggest reduced PMS2 interaction and MMR activity [44,45]. In both assays, the empty vector displayed a strong growth defect in the presence of MTX and absence of histidine, respectively, compared to the wild-type (Fig 2A). The Q542L, S406N, and I219V variants showed similar growth to the wild-type, whereas the L550P and T117M variants exhibited a similar growth defect to the empty vector (Fig 2A). This suggests that the L550P and T117M variants likely lose PMS2 interaction due to reduced cellular abundance. In conclusion, this data confirmed that the DHFR-PCA and Y2H assay could report on MLH1 abundance and its interaction with PMS2, respectively, and distinguish benign population variants from clinically pathogenic ones.

## Saturated abundance and interaction maps of MLH1 variants

Next, we used the DHFR-PCA and Y2H assay to determine the abundance and PMS2 interaction of all possible missense variants in the C-terminal domain (positions 487–756) of human MLH1 (Fig 2B). We decided to focus only on the C-terminal domain of MLH1, as it mediates the interaction with PMS2, and we expected variants in the N-terminal domain to have little impact on this interaction. We divided the C-terminal domain into seven tiles (S4A Fig) and designed site-saturation mutagenesis libraries in each tile, which were synthesized and cloned into a Gateway-compatible vector to produce the entry libraries. The entry libraries were cloned into destination vectors tailored to the DHFR-PCA and Y2H assay, respectively, and subsequently transformed into appropriate yeast strains. The transformation cultures were grown until saturation and then cells were plated in triplicates onto control and selection medium, and incubated at 30 °C. In the DHFR-PCA, cells were grown on medium with MTX to select for MLH1 abundance, and in the Y2H assay, cells were grown on medium without histidine to select for PMS2 interaction. Following incubation, cells were scraped off the plates and directly used for plasmid isolation and amplicon preparation. The amplicons were sequenced using paired-end next-generation sequencing and the resulting sequencing data was analyzed using Enrich2 [46] to calculate an abundance score and a PMS2 interaction score for each MLH1 variant (Fig 2B). The variant scores were calculated by comparing the change in frequency of each variant to that of synonymous wild-type variants in both the control and selection conditions. The MLH1 abundance scores displayed a bimodal distribution with many variants centered around 0 (wild-type-like) and a peak of low-abundance variants with negative scores (S4B Fig). However, due to the observed variance in negative scores across tiles within the low-abundance peak (S4B Fig) we rescaled the abundance scores to range from −1–1 (Fig 2C), which standardized the abundance scores and ensured uniformity across tiles. The distribution of interaction scores largely appeared centered around 0 (wild-type-like) with some variants showing reduced interaction with negative interaction scores, while a few variants exhibited improved interaction with positive interaction scores (Figs 2C, S4C). The interaction scores were not rescaled. As expected, the L550P variant displayed reduced abundance and interaction, in line with the results from the small-scale yeast growth assays and underscoring the validity of the high-throughput screens (Fig 2A, 2C).

The abundance and interaction datasets contained variant effects for 4839 (94%) out of 5170 (19 substitutions/position*270 residues + 40 nonsense mutations) possible programmed MLH1 variants in the C-terminal domain. The abundance and interaction scores showed high correlation between the three replicates, but the DHFR-PCA had relatively

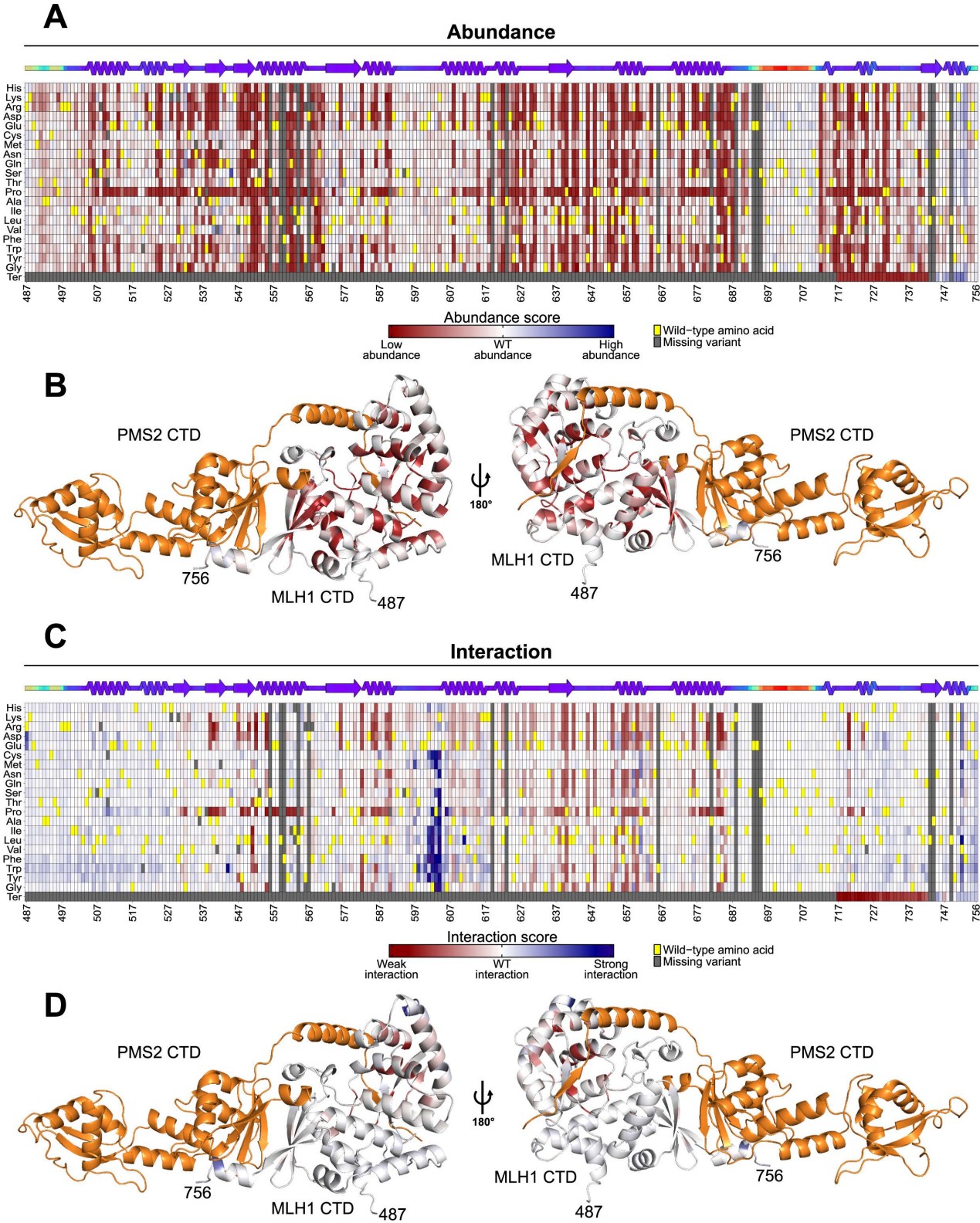

**Fig 3. MLH1 abundance and interaction heatmaps.** (A) Heatmaps showing the abundance scores of MLH1 variants. The scores range from low abundance (red), through wild-type-like abundance (white), to high abundance (blue). The wild-type residue at each position is marked in yellow, while

missing variants are marked in gray. Substitutions that lead to a stop codon (nonsense mutations) are indicted by 'Ter'. The position in the full-length MLH1 protein is shown below the heatmap. A linear representation of the secondary structure of MLH1 is displayed above the heatmap, colored by AlphaFold2 pLDDT as a measure of disorder. (B) The per-position median abundance scores are mapped to the C-terminal domain of MLH1 in the AlphaFold2-Multimer predicted C-terminal MutLα complex. The color scale corresponds to that shown in Fig 3A. The C-terminal domain of PMS2 is shown in orange. The N- and C-terminus of the C-terminal domain of MLH1 are annotated with the position in the full-length MLH1 protein. (C) Same as in (A), but for interaction scores. (D) Same as in (B), but for interaction scores.

lower correlations compared to the Y2H assay (minimum R = 0.92 vs. minimum R = 0.96) (S5, S6 Figs). Overall, substitutions in the C-terminal domain of MLH1 more often conferred reduced abundance than affecting the interaction with PMS2 (Fig 3A, 3C). The abundance map revealed that substitutions were detrimental in several positions and particularly in secondary structural elements (Fig 3A). The internal loop spanning position 591–605, as well as some positions close to the N- and C-termini were tolerant to substitutions (Fig 3A). As expected, the disordered region with low AlphaFold2-Multimer-predicted pLDDT scores at position 692–711 was insensitive to substitutions, whereas substitutions to proline appeared detrimental to abundance, and to a lesser extent to interaction, at most positions (Fig 3). Interestingly, nonsense mutations caused reduced abundance and interaction up until position 742, but from position 745–756, stop codon variants displayed increased abundance compared to the wild-type (Fig 3). Additionally, several point mutations at positions toward the C-terminus led to increased abundance and interaction (Fig 3). We mapped the median abundance scores onto the C-terminal domain of MLH1 in the AlphaFold2-Multimer-predicted structure of the human C-terminal MutLα complex (Fig 3B). This revealed that most of the sensitive positions were pointing into the core of the protein, while exposed and flexible positions were tolerant to substitutions. Indeed, the median abundance scores correlated with relative solvent accessible surface area (rSASA), and indicated that substitutions in buried, core positions were more detrimental than in surface-exposed positions (S7A Fig).

The interaction map revealed that the impact of MLH1 variants on the interaction with PMS2 was confined to specific positions rather than being widespread (Fig 3C). In general, most variants that displayed reduced interaction with PMS2 also appeared low-abundant, suggesting that loss of stability was the primary mechanism of reduced interaction with PMS2 (Fig 3A, 3C). In line with this, we observed that low median interaction scores were only observed for buried positions (S7B Fig). Large regions at the beginning and end of the C-terminal domain of MLH1 were completely tolerant to substitutions, with most variants exhibiting wild-type-like interaction with PMS2 (Fig 3C). It was evident that substitutions from hydrophobic wild-type residues to both negatively and positively charged amino acids in the β-strands constituting the canonical interface at position 539–541 and 547–550 were detrimental to the interaction (Fig 3C). Interestingly, at position 540, the substitution to glutamic acid had no effect on the interaction with PMS2, whereas the substitution to aspartic acid displayed complete loss-of-interaction (Fig 3C). We noticed that the introduction of tryptophan at position 544, which is located in the short loop between the second and third β-strand of the canonical interface, substantially improved the binding with PMS2 (Fig 3C). In addition, MLH1 variants immediately upstream of the central α-helix, specifically at position 601–604, showed increased binding with PMS2 (Fig 3C). This effect was most pronounced for substitutions to hydrophobic or aromatic residues and disappeared with substitutions to charged residues (Fig 3C). To validate the effects on PMS2 interaction observed in this screen, we performed low-throughput Y2H growth assays on four loss-of-interaction variants and three gain-of-interaction variants. These experiments confirmed the effects observed in the high-throughput screen (S8 Fig).

To visualize the variant effects on PMS2 interaction, we mapped the median interaction scores onto MLH1 in the predicted C-terminal MutLα complex (Fig 3D). The mapping revealed that substitutions of surface-exposed residues in the central α-helix spanning position 605–627 exhibited slightly decreased interaction with PMS2, without affecting the MLH1 abundance (Fig 3B, 3D). This led us to speculate that MLH1 and PMS2 interact through an additional interface, distal to the canonical four β-strands known from the yeast complex [35] (S9 Fig). Indeed, when we predicted the full structure of

the human MutLα complex using AlphaFold 3, we identified an α-helix (positions 409–431) within the intrinsically disordered linker region of PMS2 that directly interacts with the distal central α-helix in MLH1 (S10A, S10B Fig). Furthermore, mapping the median interaction scores to the structure revealed that several contact residues in MLH1 within 6 Å of the PMS2 409–431 α-helix located to positions with both loss- and gain-of-interaction effects in our Y2H assay (S10C Fig). These findings support the additional interface in the AlphaFold 3-predicted model and suggest that the disordered linker region in PMS2 harbors an α-helix forming a previously uncharacterized interface with MLH1.

The mapping of the median interaction scores to the structure also revealed that several positions in the helices neighboring the central α-helix in MLH1 were highly sensitive to substitutions (Fig 3D). However, these residues pointed into the protein core, and substitutions also resulted in reduced abundance, suggesting that the loss of interaction was likely due to decreased stability (Fig 3B, 3D).

## Computational predictions and clinical data correlate with variant effects

The above data indicate that MLH1 abundance and its interaction with PMS2 are tightly connected and loss of interaction is often caused by loss of stability. To further examine this relationship, we used predictive computational tools to evaluate the impact of MLH1 variants on thermodynamic folding stability and their likelihood of being tolerated based on the evolutionary history. We used Rosetta [47] to predict the changes in thermodynamic stability (ΔΔG) of all possible missense variants in the C-terminal domain of MLH1. However, in addition to stability, certain residues may be important for other aspects of MLH1 function, such as its interaction with PMS2. Such residues are often conserved across species, and therefore analysis of homologous sequences can report on the mutational tolerance at these positions. To this end, we used GEMME [48] to model the evolutionary conservation of MLH1 sequences and then predict the required adaptation of the MLH1 sequence to accommodate all possible missense variants in the C-terminal domain of MLH1. GEMME returned a ΔΔE score for each MLH1 variant, ranging from 0 (tolerated substitution) to -6 (non-tolerated substitution based on evolutionary conservation). Rosetta and GEMME separated the MLH1 variants according to their predicted effects on MLH1 stability and other properties important for function, respectively (Figs 4A, S11, S12). We observed that the majority of variants were predicted to have a ΔΔG score below 2 and ΔΔE score above -2, suggesting that they exhibited similar stability and functional properties as the wild-type (Fig 4A). Several variants were predicted to be incompatible with MLH1 function (strongly negative ΔΔE scores) and some of them were also predicted to be unstable (positive ΔΔG score) (Fig 4A). We mapped the experimental abundance scores to the predictions, which revealed that variants with wild-type-like abundance clustered in the upper left corner, whereas low-abundant MLH1 variants mapped towards the bottom right corner (Fig 4A). An analogous pattern appeared when we mapped the experimental interaction scores to the predictions (Figs 4B, S13). This indicates that most of the loss-of-interaction variants were predicted to be detrimental based on the evolutionary history while also losing structural stability and displaying reduced abundance levels, confirming the close relationship between MLH1 abundance and its interaction with PMS2.

To assess the clinical relevance of the generated datasets, we next compared the experimental abundance and interaction scores with the clinical classifications in the ClinVar database [41] and population allele frequencies of known MLH1 variants that have been reported in gnomAD [42] (Fig 4C, 4D). This showed that variants that are common in the human population tended to behave like the wild-type, whereas several of the rare population MLH1 variants displayed reduced abundance and interaction with PMS2. To evaluate if our abundance and interaction scores could differentiate harmless benign variants from disease-associated pathogenic variants, we colored the MLH1 variants based on their classification in ClinVar [41] (Fig 4C, 4D). The five pathogenic MLH1 variants were extremely rare in the population, and all displayed much reduced abundance (Fig 4C). Conversely, most of the benign and likely benign MLH1 variants were frequently found among individuals and exhibited wild-type-like protein levels (Fig 4C). Additionally, several rare VUS exhibited similarly low abundance levels as the pathogenic variants, indicating that these variants are likely to be disease-causing (Fig 4C). The K618T variant had an abundance score of −1, despite being classified as benign (Fig 4C). This variant has previously

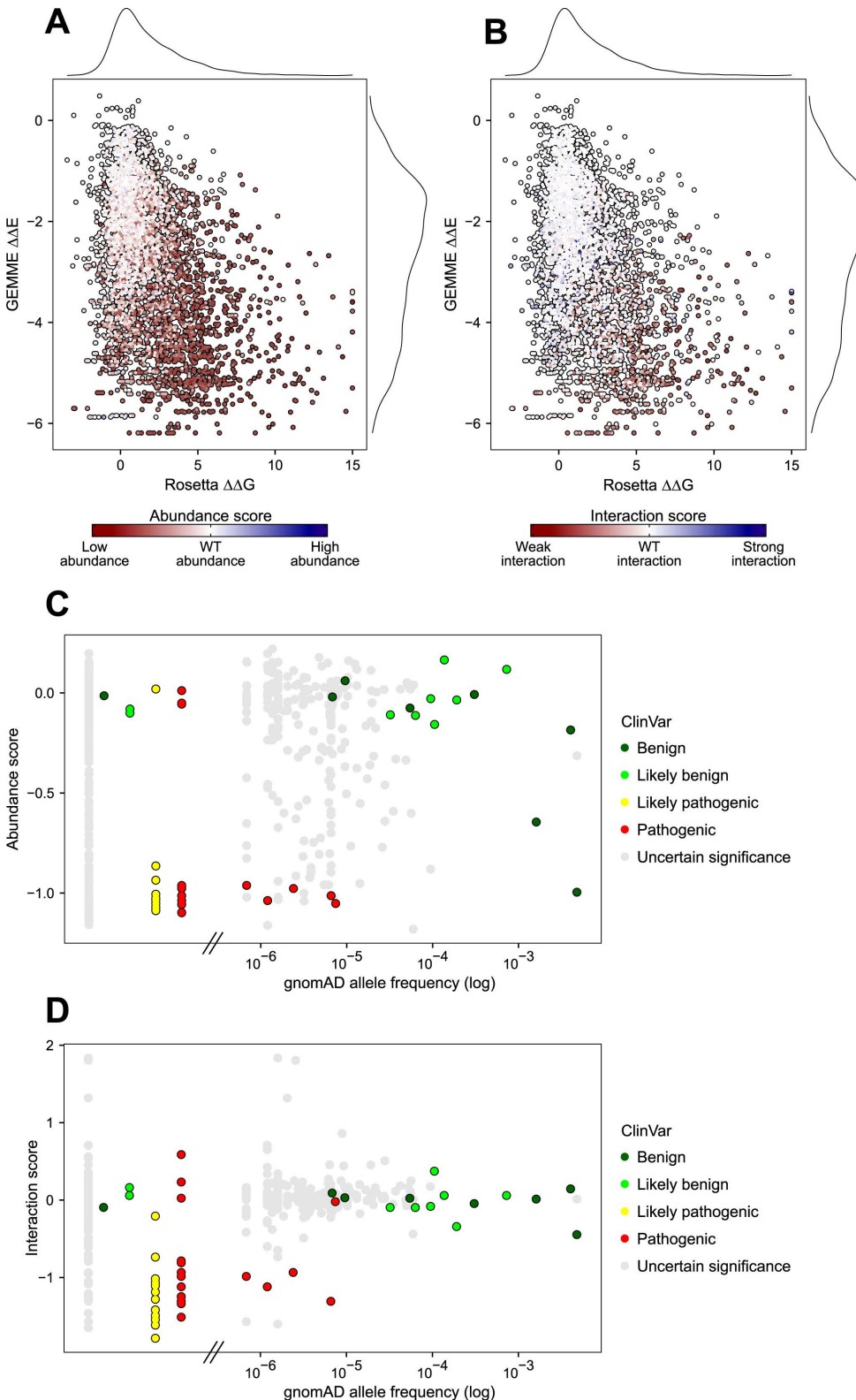

**Fig 4. *Experimental effects correlate with computational predictions and clinical classifications.*** (A) Scatter plots showing the correlation between the predicted GEMME ΔΔE score and predicted Rosetta ΔΔG score. A negative ΔΔE score suggests that the variant may not be compatible

with MLH1 function as encoded in the sequence family. A positive ΔΔG score suggests that the variant is less stable than the wild-type MLH1. The MLH1 variants are colored by the abundance score and the color scale corresponds to that shown in Fig 3A. (B) Same as in (A), but for interaction scores. (C) Scatter plots showing the correlation between the abundance score and the gnomAD allele frequency (log scale). The MLH1 variants are colored based on their classification in ClinVar as either benign (dark green), likely benign (light green), likely pathogenic (yellow), pathogenic (red) or uncertain significance (gray). Variants observed in patients but not in gnomAD are indicated to the left of the broken x-axis. These variants were retrieved from Simple ClinVar [34] (accessed July 25, 2024). (D) Same as in (C), but for interaction scores.

been shown to have reduced cellular abundance, nuclear localization and MMR activity [43,49–52], while other studies have suggested this variant to form a stable protein [31]. Thus, despite the variant being too frequent in the population to be a fully penetrant Lynch syndrome variant, its clinical interpretation remains inconclusive [53]. The interaction scores were also able to distinguish most benign from pathogenic variants (Fig 4D). The pathogenic R687T variant showed wild-type-like interaction with PMS2 but dramatically decreased abundance, suggesting that the disease-mechanism of this variant is through loss of stability (Fig 4C, 4D). Interestingly, some of the MLH1 variants that displayed improved interaction with PMS2 were classified as VUS but were extremely rare in the population (Fig 4D). This could suggest that these variants are pathogenic, and thus gain of interaction may represent an additional underlying disease-mechanism of Lynch syndrome. Lastly, we performed receiver operating characteristic (ROC) analyses and calculated the area under the curve (AUC) to assess whether our experimental scores and computational predictions could distinguish pathogenic variants from benign ones (S14 Fig). This analysis revealed that both our abundance and interaction scores accurately predict disease-causing MLH1 variants, both achieving AUCs of 0.89 (S14 Fig). Similarly, Rosetta (AUC = 0.84) and GEMME (AUC = 0.91) demonstrated comparable predictive power, suggesting that these computational prediction models capture MLH1 pathogenicity almost as effectively as our experimental data (S14 Fig). However, it is important to emphasize that although GEMME slightly outperforms our experimental data, it only generally predicts MLH1 function and does not provide mechanistic insights into variant effects. It should also be noted that the test set only consists of 46 clinically classified variants, so additional data might reveal difference in performance.

## Discussion

Most proteins have several properties that contribute to their function. Here, we have combined two multiplexed assays of variant effects (MAVEs), one probing stability and another probing interaction, to disentangle which variants affect either or both properties in the C-terminal domain of human MLH1.

Loss-of-function mutations in the *MLH1* gene lead to Lynch syndrome, a cancer predisposing disease. Loss of stability has been shown to be a common mechanism of loss-of-function phenotypes [3,31,32,54,55], and pathogenic variants are enriched in interaction interfaces [17–20]. Our data helps pinpoint whether the molecular mechanism underlying pathogenicity of a given variant is loss of stability, loss of interaction with PMS2, or yet other properties of the protein. Our results indicate that several mutations reduce protein stability, while far fewer variants affect the interaction. At the same time, variants that result in a loss of interaction often display reduced abundance levels. These observations suggest that it is difficult to perturb the interaction with PMS2 without also compromising the structure of the MLH1 protein. We envision that this intricate relationship between structural protein stability and protein-protein interactions (PPIs) reflects the nature of large and structured interfaces, such as the interaction between MLH1 and PMS2. Thus, for single MLH1 missense mutations to affect the interaction with PMS2, it requires strong effects that are unlikely to occur without also structurally affecting MLH1 stability. This contrasts with PPIs mediated through smaller interfaces and involving unstructured components, such as short linear motifs (SLiMs) in intrinsically disordered regions [56,57]. In those cases, missense mutations are expected to have local effects on specific binding sites rather than globally perturbing protein stability. We speculate that the novel interaction between the α-helix in PMS2 (positions 409–431) and the distal central α-helix in the C-terminal domain of MLH1 represents such a motif-mediated interaction (S10 Fig), which explains why individual substitutions specifically disrupt the interaction without compromising protein stability.

Despite the difficulty in disentangling protein stability and PMS2 interaction for MLH1 variants, we observe that substitutions to charged residues in two of the four β-strands constituting the canonical interface with PMS2 are detrimental to the interaction. We note that this interface is enriched with hydrophobic residues and completely depleted of charged residues, which may explain why the introduction of charged residues in these β-strands is unfavorable for the interaction with PMS2 (S15 Fig) [58]. Interestingly, at position 540, only substitutions to aspartic acid and arginine are detrimental, whereas substitutions to glutamic acid and lysine exhibit wild-type-like interaction. This suggests that charge is not the only determinant of the effect on PMS2 interaction; spatial constraints and the local environment of the interface also influence the effect of substitutions. We identify a single gain-of-interaction variant, Q544W, in the canonical interface, located in the short loop between the second and third β-strand. We speculate that introducing a tryptophan residue at this site might alter the loop's conformation, and that the inherent stickiness of tryptophan could promote new molecular interactions with PMS2. In addition to the canonical interface, substitutions within and immediately before a central α-helix of MLH1 lead to reduced and increased binding with PMS2, respectively. This effect is presumed to occur via an α-helix that extends towards the disordered linker region of PMS2. We hypothesize that this is a novel interaction site between MLH1 and PMS2.

Counterintuitively, some MLH1 variants that appear unstable still exhibit wild-type-like interaction with PMS2. We confirm this observation for selected variants in low-throughput (S16A, S16B Figs). This suggests differences in the dynamic range and sensitivity of the DHFR-PCA and Y2H assay. We believe there are several possible explanations for this. First, the expression of the *HIS3* gene in the Y2H assay likely provides a much more sensitive read-out compared to the reconstitution of the murine DHFR in the DHFR-PCA. This sensitivity difference could potentially be mitigated by using a higher concentration of 3-AT to increase the selection pressure on the interaction, thereby shifting the dynamic range to select for variants that mildly affect the interaction without perturbing the protein stability. Second, it has been shown that the DHFR-PCA underestimates the abundance of proteins localized to subcellular compartments (e.g., the nucleus), as the binding of the F[1,2] and F[3] fragments of murine DHFR occurs in the cytosol [38]. However, studies have shown that translocation of MLH1 into the nucleus depends on its heterodimerization with PMS2 [59–62]. Since the DHFR-PCA is performed without co-expression of PMS2, a large fraction of MLH1 is expected to be present in the cytosol. We hypothesize that this ectopic cytosolic localization and absence of PMS2 negatively affect MLH1 stability. Despite these limitations, we observe a reasonable correlation with existing literature on MLH1 abundance [31,63] and PMS2 interaction [44], especially given differences in cell types and experimental techniques used (S17 Fig).

Our abundance and interaction maps reveal that removal of residues 745–756 of MLH1 leads to both enhanced stability and interaction with PMS2. We find it unlikely that this effect is due to a decrease in degron potency, as this region is not enriched with hydrophobic residues nor depleted in acidic residues, which are characteristic of protein quality control degrons [64–66]. Additionally, since the region near the C-terminus does not contain any known C-degrons [67–70], deleting these residues should not lead to increased abundance by removing a C-degron. However, the C-terminus of MLH1 contains an extremely conserved FERC motif, which is suggested to have important regulatory and activity functions [35,71–74]. Therefore, we hypothesize that truncating the C-terminus stabilizes the protein and creates additional space around the canonical interface, thereby enhancing binding with PMS2, while potentially affecting other properties of MLH1 function mediated by the FERC motif. In general, our experimental abundance and interaction effects are in line with clinical variant classification, as well as with thermodynamic stability predictions and evolutionary conservation. It is, however, important to emphasize that this is not a symmetric relationship. Our assays can identify residues that are detrimentally affected and hence at high risk of being pathogenic. If a variant behaves like wild-type in both assays, though, this does not necessarily imply that these variants have functional MMR activity or should be considered benign. As an example, beyond the truncated C-terminus, some wild-type-like variants are predicted to be extremely rare or absent from the family of MLH1 homologs ($\Delta\Delta E < -3$). These variants could be in positions that are critical for other functions of the C-terminal domain of MLH1, such as its interaction with EXO1, which occurs via a different binding site than its interaction with PMS2

[35,75]. It has also been observed that mutational consequences may be coupled to other properties than the interaction partner binding to the site of mutation [76] and it will be interesting to revisit this when data on additional MLH1 binding partners is available.

In conclusion, our data suggests that the abundance of MLH1 and its interaction with PMS2 are closely related, with impaired interaction often caused by reduced cellular abundance. We identify detrimental mutations in the canonical interface with PMS2 and uncover a potential distal binding site between MLH1 and PMS2. Our abundance and interaction maps may aid in interpreting clinically observed MLH1 variants and enhance our understanding of their effects in Lynch syndrome.

## Materials and methods

### Plasmids and cloning

The DNA sequence of human wild-type MLH1 (UniProt ID: P40692) and human wild-type PMS2 (UniProt ID: P54278) were codon-optimized for yeast expression and cloned into pDONR221 (Genscript). Missense variants were created by Genscript. For the Y2H assay, entry clones were cloned into pDEST22 (prey) or pDEST32 (bait) from the ProQuest Two-Hybrid System (Invitrogen) using Gateway cloning (Invitrogen). For the DHFR-PCA, entry clones were cloned into pDEST-DHFR-PCA using Gateway cloning (Invitrogen).

### Yeast strains and media

The Y2H assay was performed in the MaV203 yeast strain from the ProQuest Two-Hybrid System (Invitrogen) and the DHFR-PCA was performed in the BY4741 wild-type yeast strain. Single variant transformations were done as described in [77]. Yeast cells were grown in yeast extract, peptone, dextrose (YPD) medium (2% glucose, 2% tryptone, 1% yeast extract) and in synthetic complete (SC) medium (2% glucose, 0.67% yeast nitrogen base without amino acids (Sigma), 0.2% synthetic drop-out supplement (Sigma)). For the Y2H assay, SC medium without histidine was prepared with 20 mM 3-amino-1,2,4-triazole (3-AT) (Sigma-Aldrich), unless otherwise specified. For the DHFR-PCA yeast growth assays, SC medium without uracil was prepared using drop-out supplement. For the DHFR-PCA library screen, SC medium without uracil was prepared using 76 mg/L histidine, 380 mg/L leucine, 76 mg/L methionine. Selection medium for the DHFR-PCA was prepared using 200 µM methotrexate (MTX) (Sigma-Aldrich) and 1 mM sulfanilamide (Sigma-Aldrich). Control medium for the DHFR-PCA was prepared using a corresponding volume of DMSO. Solid media was prepared using 2% agar.

### Yeast growth assays

For growth assays on solid media, cultures were grown overnight (170 rpm, 30 °C) to exponential phase. The cultures were harvested and washed in sterile water (3000 rpm, 5 minutes, room temperature) and the $OD_{600nm}$ was adjusted to 0.4. Then, cultures were used for five-fold dilution series in sterile water and 5 µL of each dilution was spotted onto agar plates. The plates were briefly air-dried before they were incubated at 30 °C for 2–3 days.

### Library design and cloning

Full-length human MLH1 was divided into seven tiles of 28, 40 or 42 residues. The residues included in each tile were: tile 1: 487–526, tile 2: 527–568, tile 3: 569–596, tile 4: 597–636, tile 5: 637–676, tile 6: 677–716 and tile 7: 717–756. For each tile, the library was synthesized and cloned into pDONR221 by Twist Bioscience. The libraries comprised all possible single amino acid substitutions at each position as well as the wild-type sequence and a synonymous wild-type mutation at each position. Tile 7 also included a nonsense mutation at each position. Mutagenesis of 15 positions and a total of 307 variants failed during library preparation by Twist Bioscience. The final libraries comprised 4839 out of 5170 (94%)

designed possible single substitution variants and nonsense variants (19 substitutions/per residue*270 residues + 40 nonsense variants). From this point forward, libraries in each tile were handled individually. Entry libraries were cloned into appropriate destination vectors using Gateway cloning (Invitrogen). Briefly, LR reactions were prepared with 243 ng (for the Y2H assay) or 260 ng (for the DHFR-PCA) pDONR221-library, 450 ng pDEST22 (for the Y2H assay) or 450 ng pDEST-DHFR-PCA (for the DHFR-PCA), 6 µL Gateway LR Clonase II enzyme mix (ThermoFisher) and TE buffer to 30 µL. The LR reactions were incubated overnight at room temperature and the following day, the reactions were stopped by adding 3 µL proteinase K (10 minutes, 37 °C). The cloned libraries were electroporated into NEB 10-beta electrocompetent *E. coli* cells (New England Biolabs) using 4 µL LR reaction and 100 µL cells at 1.8 kV. Cells were recovered in 3.9 mL NEB 10-beta stable outgrowth medium for 1 hour (250 rpm, 37 °C). Then, cells were transferred to 36 mL LB medium and 4 µL was plated on LB medium with ampicillin and incubated overnight (37 °C) for estimating the transformation efficiency. The remaining cells were plated on large BioAssay plates (500 cm$^2$) containing LB medium with ampicillin and incubated overnight (37 °C). For each tile, the number of single colonies exceeded the expected number of variants by at least 100-fold. The following day, cells were scraped off the plates with sterile water and library DNA was isolated and purified using the NucleoBond Xtra Midi kit (Macherey-Nagel).

## Library transformation and selection

The cloned MLH1 libraries in each tile were transformed and plated for selection separately. The libraries cloned into pDEST22 were transformed into the MaV203 yeast strain with pDEST32-PMS2 to perform the Y2H assay. The libraries cloned into pDEST-DHFR-PCA were transformed into the BY4741 wild-type yeast strain to perform the DHFR-PCA. Transformations were done using 30 × scale-up as described in [78]. For the Y2H assay, the MaV203 yeast strain with pDEST32-PMS2 was grown overnight (170 rpm, 30 °C) in SC-leucine medium. For the DHFR-PCA, the BY4741 wild-type yeast strain was grown overnight (170 rpm, 30 °C) in YPD medium. Upon reaching late-exponential phase, the cultures were diluted in at least 150 mL SC-leucine medium (for the Y2H assay) or YPD medium (for the DHFR-PCA) to an OD$_{600nm}$ of 0.3 and incubated (170 rpm, 30 °C) until an OD$_{600nm}$ of 1.2 was reached. The cells were harvested and washed in sterile water (3000 rpm, 5 minutes, room temperature). The cell pellets were resuspended in a transformation mix and incubated in a water bath (40 minutes, 42 °C). The transformation mixed contained: 7.2 mL 50% PEG-3350 (Sigma), 1.08 mL 1 M LiAc (Sigma), 300 µL 10 mg/mL freshly denatured carrier DNA (Sigma), 30 µg cloned library plasmid DNA and sterile water to 10.8 mL. Then, the cells were harvested (3000 rpm, 5 minutes, room temperature) and the transformation mix was discarded. The cell pellets were resuspended in 30 mL sterile water and 5 µL and 10 µL were plated on SC-leucine-tryptophan medium (for the Y2H assay) or SC-uracil medium (for the DHFR-PCA) and incubated for two days (30 °C) to evaluate the efficiency of the transformations. The remaining cells were diluted in SC-leucine-tryptophan medium (for the Y2H assay) or SC-uracil medium (for the DHFR-PCA) to an OD$_{600nm}$ of 0.2 and incubated for two days (170 rpm, 30 °C). For each tile, for both the DHFR-PCA and Y2H assay, every variant was represented at least 100 times. After two days, the transformation cultures reached saturation. For the Y2H assay, 2 × 4.5 OD$_{600nm}$ units were harvested (17,000 g, 1 minute, room temperature) in triplicates, washed in sterile water and resuspended in 600 µL sterile water. Then, 300 µL was plated on large BioAssay plates (500 cm$^2$) containing SC-leucine-tryptophan medium to serve as the control condition. The remaining 300 µL was plated on large BioAssay plates (500 cm$^2$) containing SC-leucine-tryptophan-histidine medium with 20 mM 3-amino-1,2,4-triazole (3-AT) to serve as the selection condition. The plates were air-dried and incubated at 30 °C for two days. For the DHFR-PCA, 2 × 22.5 OD$_{600nm}$ units were harvested (17,000 g, 1 minute, room temperature) in triplicates, washed in sterile water and resuspended in 600 µL sterile water. Then, 300 µL was plated on large BioAssay plates (500 cm$^2$) containing SC+histidine+leucine+methionine medium to serve as the control condition. The remaining 300 µL was plated on large BioAssay plates (500 cm$^2$) containing SC+histidine+leucine+methionine medium with 200 µM MTX and 1 mM sulfanilamide to serve as the selection condition. The plates were air-dried and incubated at 30 °C for three days. Following the incubation, yeast cells were scraped off the plates using 30 mL sterile water and 9 OD$_{600nm}$ units from

each plate was harvested (17,000 g, 1 minute, room temperature) and used for plasmid purification. Library plasmid DNA was isolated from yeast cells using the ChargeSwitch Plasmid Yeast Mini kit (Invitrogen).

## Amplicon preparation and library sequencing

Amplicons for each tile were prepared via two rounds of PCR, with the product cleaned-up after each reaction. For the first PCR, 50 μL reactions were prepared by mixing 25 μL of 2 × Q5 High-Fidelity PCR Master Mix (New England Biolabs), 5 ng of purified library plasmid DNA, 2.5 μL of 10 μM forward and reverse tile-specific primers with Illumina adapter sequences, and nuclease-free water to reach a total volume of 50 μL. The primer sequences and annealing temperatures are listed in the supplementary material (S1 File). The reactions were subjected to the following PCR program: 98 °C for 30 seconds, followed by 8 cycles of 98 °C for 10 seconds, 57–64 °C for 30 seconds, 72 °C for 20 seconds, followed by 72 °C for 2 minutes and 4 °C hold. The product from the first PCR was cleaned-up using Ampure XP beads (Beckman Coulter). Briefly, 15 μL of the PCR product were thoroughly mixed with 14.25 μL Ampure XP beads in 1.5 mL Eppendorf tubes and incubated at room temperature for 5 minutes. Following the incubation, the tubes were placed on a magnetic stand and the supernatant was carefully removed before the beads were washed twice with 200 μL 70% ethanol. The beads were air-dried at room temperature for 15 minutes to allow residual ethanol to evaporate. Finally, the cleaned-up PCR product was eluted from the beads with nuclease-free water and mixed thoroughly. The tubes were placed on a magnetic stand and the cleared supernatant was transferred to a fresh 1.5 mL Eppendorf tube. Then, 50 μL reactions were prepared for the second PCR by mixing 25 μL of 2 × Q5 High-Fidelity PCR Master Mix (New England Biolabs), 15 μL of cleaned-up PCR product from the first PCR and 10 μL Nextera DNA CD Indexes (96-well format from Illumina) for dual indexing. The reactions were subjected to the following PCR program: 98 °C for 30 seconds, followed by 12 cycles of 98 °C for 10 seconds, 62 °C for 30 seconds, 72 °C for 20 seconds, followed by 72 °C for 2 minutes and 4 °C hold. The product from the second PCR was cleaned-up as described for the first PCR, using 11.25 μL Ampure XP beads. Finally, the concentration of the amplicons was measured using the Qubit 2.0 Fluorometer (Invitrogen) with the Qubit dsDNA HS Assay kit (Thermo Fisher Scientific) and the integrity was assessed using the 2100 Bioanalyzer (Agilent Technologies) with the High Sensitivity DNA kit. The amplicons were volumetrically pooled and diluted before they were subjected to pair-end next-generation sequencing using the Illumina NextSeq 550 System with the Mid Output v2.5 300 cycle kit (Illumina).

## Analysis of sequencing data

FASTQ files with read 1 and read 2 were downloaded from BaseSpace and sequences from the four lanes were merged. The quality and length of sequences in the merged read 1 and read 2 FASTQ files were assessed using FastQC [79]. Then, Trimmomatic [80] paired-end mode was used to obtain sequences of 151 base pairs using the following flags: MINLEN:151 and CROP:151. The FASTQ files with trimmed sequences were used as input to Enrich2 [46] to count reads and calculate a score for each variant. Enrich2 calculated log ratios and standard errors for each variant based on three replicates using the control condition as time point 1 and the selection condition as time point 2. The scores were calculated by normalization to synonymous wild-type variants. Enrich2 was run using the overlap mode with default parameters, but with a minimum and average quality of 1, maximum N's of 0, maximum mismatches of 0, and 'remove unresolvable overlaps' and 'overlap only' set to TRUE. The start and end were appropriately indicated for each tile. The wild-type sequence was included with the correct offset for each tile. For the DHFR-PCA, the scores appeared clearly bimodal with a wild-type-like peak centered around 0 and a peak of unstable variants with negative scores (S4B Fig). However, the low-abundance peak displayed varying scores between tiles. Hence, to put all tiles on a similar scale, the DHFR-PCA scores were rescaled per tile by dividing all scores with the score of the low-abundance peak, giving scores ranging from −1–1, with wild-type centered at 0. For the Y2H assay, the score distributions did not appear bimodal, but instead all tiles had a sharp wild-type-like peak centered around 0 and few loss- or gain-of-interaction variants, and therefore, we omitted rescaling the scores (S4C Fig). The heatmaps were produced in R using the packages ComplexHeatmap [81] and circlize [82].

## Modeling and structural analysis

The C-terminal MutLα complex was predicted using AlphaFold2-Multimer. AlphaFold2-Multimer was run using ColabFold [83]. The C-terminal domain of MLH1 (UniProt ID: P40692) included residue 481–756 and the C-terminal domain of PMS2 (UniProt ID: P54278) included residue 611–862. The N-terminal domains of MLH1 and PMS2 shown in Fig 1A are PDB: 4P7A [84] and 1H7S [85], respectively.

The surface hydrophobicity of the C-terminal domain of MLH1 was assessed using the YRB script [58] in PyMOL with the AlphaFold2-predicted structure. We used the predicted structure of the C-terminal domain of MLH1 due to the homomeric nature of the solved structure (PDB: 3RBN), which rendered it unsuitable for this analysis.

The full structure of the human MutLα complex was predicted using AlphaFold 3 [86] https://alphafoldserver.com. We used the best-scoring model, which is available in the GitHub repository for this project.

## Thermodynamic stability predictions

Rosetta with the Cartesian ΔΔG protocol [47] (GitHub SHA1 99d33ec59ce9fcecc5e4f3800c778a54afdf8504) was used on the crystal structure of the C-terminal domain of MLH1 (PDB: 3RBN) to predict the changes in thermodynamic stability of MLH1 variants. The MLH1 structures were prepared and relaxed and inputted for ΔΔG calculations using an in-house pipeline (https://github.com/KULL-Centre/PRISM/tree/main/software/rosetta_ddG_pipeline, v0.2.1). The change in folding stability was calculated by subtracting the wild-type ΔG value from the variant ΔG value. The ΔΔG values in Rosetta Energy Units were divided by 2.9 to convert them to kcal/mol [47].

## Evolutionary conservation predictions

The evolutionary conservation ΔΔE score was calculated using GEMME [48]. The first isoform of MLH1 (UniProt: P40692-1) was input into HHblits [87] to build an MSA of MLH1 homologs using an E-value threshold of $10^{-10}$. The resulting MSA was refined by first including only positions present in the wild-type MLH1 sequence and then by excluding sequences with more than 50% gaps. This process resulted in a final MSA containing 683 MLH1 homologs, which was input into GEMME to estimate the evolutionary distance of the MLH1 variants from the wild-type MLH1 sequence.

## Supporting information

**S1 Fig.** *AlphaFold2-Multimer prediction of the C-terminal MutLα complex.* High confidence model of the human C-terminal MutLα complex predicted by AlphaFold2-Multimer. The structure is colored by the pLDDT score using the reversed rainbow color palette in PyMOL.
(TIFF)

**S2 Fig.** *Vector map of pDEST-DHFR-PCA.* Vector map of the CEN-based expression vector pDEST-DHFR-PCA with a *URA3* marker. The mDHFR[F3]-MLH1 fusion protein and mDHFR[F1,2] were expressed from the CYC1 and GAP promoter, respectively. The vector map was generated using SnapGene.
(TIFF)

**S3 Fig.** *Vector maps of pDEST22 and pDEST32.* Vector maps of the CEN-based expression vectors pDEST22 and pDEST32 with a *TRP1* and *LEU2* marker, respectively. MLH1 was fused to the activation domain and PMS2 was fused to the DNA binding domain. The vector maps were generated using SnapGene.
(TIFF)

**S4 Fig.** *Distributions of abundance and interaction scores per tile.* (A) The predicted structure of the C-terminal MutLα complex with the C-terminal domain (CTD) of MLH1 colored by the seven tiles. The tiles are colored according to the legend in (C). The C-terminal domain of PMS2 is shown in orange. (B) Density plot showing the distribution of

abundance scores for each tile in the C-terminal domain of MLH1 prior to rescaling. The tiles are colored according to the legend in (C). (C) Same as in (B), but for interaction scores.
(TIFF)

**S5 Fig.** *Score correlation between replicates.* Scatter plots showing the score correlation between the three replicates for DHFR-PCA (abundance) and Y2H (interaction). The Pearson correlation coefficient (R) is shown for each plot.
(TIFF)

**S6 Fig.** *Heatmaps of the standard error (SE) of abundance and interaction scores.* (A) Heatmaps of the SE for the abundance scores, calculated using Enrich2. The SE has been rescaled in accordance with the rescaling of the abundance scores. The SE range from 0 (white) to 0.45 (dark green). The wild-type residue at each position is marked in yellow, while missing variants are marked in gray. The position in the full-length MLH1 protein is shown below the heatmap. A linear representation of the secondary structure of MLH1 is displayed above the heatmap, colored by AlphaFold2 pLDDT as a measure of disorder. (B) Same as in (A), but for interaction scores. Note, the color scale is different from in (A).
(TIFF)

**S7 Fig.** *Median score correlation with relative solvent accessible surface area (rSASA).* (A) Scatter plot showing the rSASA plotted against the median abundance score per position. Residues are colored by the secondary structural element according to the legend in (B). A rSASA value of 0 corresponds to a completely buried residue, while 1 corresponds to a completely exposed residue. The Spearman's R correlation coefficient is shown. (B) Same as in (A), but for median interaction scores.
(TIFF)

**S8 Fig.** *Validation of loss- and gain-of-interaction variants.* (A) Scatter density plot showing the interaction score plotted against the abundance score. The color scale is indicated in the legend. Selected loss-of-interaction and gain-of-interaction variants are highlighted in red. (B) The structure of the predicted C-terminal MutLα complex with the selected MLH1 variants highlighted. (C) Y2H growth assays comparing the growth of the vector, wild-type MLH1 and the indicated MLH1 variant. Cells were grown on medium with histidine and without histidine with 20 mM (the concentration used in the high-throughput screen), 40 mM or 60 mM 3-AT.
(TIFF)

**S9 Fig.** *Interfaces of the MutLα complex.* The AlphaFold2-Multimer predicted structure of the human C-terminal MutLα complex shows the interaction between MLH1 (purple) and PMS2 (orange). The canonical interface known from the yeast complex and the novel interaction site mediated via the distal central α-helix are indicated by black arrows.
(TIFF)

**S10 Fig.** *Prediction of a novel interaction site with PMS2.* (A) The full structure of the human MutLα complex as predicted by AlphaFold 3. As in other figures, MLH1 is colored purple, and PMS2 is colored orange. The intrinsically disordered linker regions between the N- and C-terminal domains are colored magenta and yellow, respectively. The N-terminal domains and linker regions are displayed as transparent. AlphaFold 3 predicts an α-helix within the disordered linker of PMS2, spanning residues 409–431 (yellow), which binds to the distal central α-helix in MLH1. (B) A zoomed-in view of the interaction between the α-helix in PMS2 (yellow) and the central α-helix in MLH1 (purple). Residues facing MLH1 in the PMS2 α-helix are displayed as sticks, with nitrogen atoms in sidechains highlighted in blue. (C) The same view as in (B), with MLH1 colored by the per-position median interaction score. The color scale corresponds to that shown in Fig 3C. Contact residues in MLH1 within 6 Å of the PMS2 α-helix are shown as sticks. Notably, many of these residues locate to positions where our experiments showed both loss- and gain-of-interaction variants. For visual clarity, the coloring of nitrogen and oxygen atoms in the sidechains has been omitted.
(TIFF)

**S11 Fig.** *Correlation of Rosetta and GEMME predictions with experimental scores.* (A) Scatter density plot comparing the predicted Rosetta ΔΔG scores and the abundance scores. The color scale is indicated in the legend. The Spearman's R correlation coefficient is shown. (B) Same as in (A), but for predicted GEMME ΔΔE scores. (C) Scatter density plot comparing the predicted Rosetta ΔΔG scores and the interaction scores. The color scale is indicated in the legend. The Spearman's R correlation coefficient is shown. (D) Same as in (C), but for predicted GEMME ΔΔE scores.
(TIFF)

**S12 Fig.** *Predicting abundance and interaction scores using Rosetta and GEMME scores.* ROC curves to assess how well Rosetta and GEMME perform in predicting the abundance and interaction scores. A cutoff of −0.5 was applied to the abundance and interaction scores. Variants with a score below −0.5 were categorized as detrimental, while those with a score above −0.5 were categorized as wild-type-like. The AUC for each predictor is reported in the legend. The dashed diagonal line denotes performance of a random classifier. Note that variants with scores above −0.5 include both wild-type-like variants and variants with increased abundance or PMS2 interaction.
(TIFF)

**S13 Fig.** *Separation of interaction scores using GEMME predictions and abundance scores.* Scatter plots showing the correlation between the predicted GEMME ΔΔE score and abundance scores. The MLH1 variants are colored by the interaction score and the color scale corresponds to that shown in Fig 3A.
(TIFF)\

**S14 Fig.** *Classification of MLH1 pathogenicity using experimental and computational scores.* ROC curves to assess how well our experimental abundance and interaction scores, as well as computational Rosetta and GEMME predictions perform in separating pathogenic (and likely pathogenic) from benign (and likely benign) MLH1 variants. The AUC for each predictor is reported in the legend. The dashed diagonal line denotes a random classifier.
(TIFF)

**S15 Fig.** *Hydrophobicity of the canonical PMS2 interface.* (A) Cartoon representation of the C-terminal domain of MLH1 predicted by AlphaFold2. Surface residues are colored according to their chemical properties using the YRB script (see Materials and Methods). Carbon atoms not bound to oxygen or nitrogen atoms are colored yellow (hydrophobic amino acids), negatively charged oxygen atoms are colored red (negatively charged amino acids), positively charged nitrogen atoms are colored blue (positively charged amino acids) and all other atoms are colored gray. The four β-strands in the canonical interface with PMS2 are enriched with hydrophobic amino acids (yellow) and depleted in charged residues (red and blue). (B) Same as in (A), but a different view of the structure highlighting the four β-strands at the bottom front.
(TIFF)

**S16 Fig.** *Dynamic range and sensitivity of the DHFR-PCA and Y2H assay.* (A) Scatter density plot comparing the interaction scores and the abundance scores. The color scale is indicated in the legend. Selected MLH1 variants that displayed reduced abundance but wild-type-like interaction with PMS2 are highlighted in red. (B) Yeast growth assays comparing the growth of a vector, wild-type or the indicated MLH1 variant in the DHFR-PCA (top panel) and the Y2H assay (bottom panel). In the DHFR-PCA, cells were grown on medium with or without MTX. In the Y2H assay, cells were grown on medium with or without histidine.
(TIFF)

**S17 Fig.** *Benchmarking abundance and interaction scores against existing literature.* (A) Comparison of our experimental abundance scores with protein expression levels reported in [63]. In their study, HEK293T cells were transiently transfected with an MLH1 variant, and cellular extracts were analyzed using SDS-PAGE and immunoblotting with

an anti-MLH1 antibody. Statistical analysis of MLH1 expression levels relative to WT was performed using a t-test, with p-values calculated for each variant. Variants shown in red indicate $p < 0.05$. As expected, most of the variants with significantly reduced expression levels were also measured to have reduced abundance scores in our work. (B) Comparison of our experimental abundance scores with protein expression levels reported in [31]. In their study, HCT116 cells were transiently transfected with an MLH1 variant and analyzed using immunofluorescence microscopy with an anti-MLH1 antibody. Expression levels were quantified and reported as a percentage relative to WT. As expected, most of the variants with reduced expression levels were also measured to have reduced abundance scores in our work. (C) Comparison of our experimental interaction scores with PMS2 interaction data reported in [44]. In their study, HEK293T cells were transiently co-transfected with an MLH1 variant and PMS2, and cellular extracts were analyzed using SDS-PAGE and immunoblotting with anti-MLH1 and anti-PMS2 antibodies. Based on PMS2 expression levels, MLH1 variants were classified as either interacting or not interacting with PMS2. Variants with expression levels too low to measure PMS2 expression were denoted as NA. As expected, most MLH1 variants that interacted with PMS2 in their study also exhibited positive interaction scores or scores close to 0 in our work. Additionally, variants reported in [44] to have dramatically reduced expression levels also displayed reduced interaction scores in our work.
(TIFF)

**S1 File.  All data files combined into a single Excel file**.
(XLSX)

## Acknowledgments

We acknowledge the use of the sequencing and computing core facilities at the Department of Biology, University of Copenhagen. We thank Vasileios Voutsinos for assistance with Illumina sequencing, Matteo Cagiada for running Rosetta and GEMME predictions, and members of the Linderstrøm-Lang Centre for insightful discussions. pDEST-DHFR-PCA was a gift from Professor Rasmus Hartmann-Petersen (University of Copenhagen).

## Author contributions

**Conceptualization:** Sven Larsen-Ledet, Amelie Stein.

**Data curation:** Sven Larsen-Ledet.

**Formal analysis:** Sven Larsen-Ledet, Aleksandra Panfilova, Amelie Stein.

**Funding acquisition:** Amelie Stein.

**Investigation:** Sven Larsen-Ledet.

**Methodology:** Sven Larsen-Ledet.

**Project administration:** Sven Larsen-Ledet, Amelie Stein.

**Supervision:** Amelie Stein.

**Validation:** Sven Larsen-Ledet.

**Visualization:** Sven Larsen-Ledet.

**Writing – original draft:** Sven Larsen-Ledet.

**Writing – review & editing:** Sven Larsen-Ledet, Amelie Stein.

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
