## [Decision Letter · Decision Letter 0]

1 Dec 2024

PGENETICS-D-24-01287

Disentangling the mutational effects on protein stability and interaction of human MLH1

PLOS Genetics

Dear Dr. Stein,

Thank you for submitting your manuscript to PLOS Genetics. After careful consideration, we feel that it has merit but does not fully meet PLOS Genetics's publication criteria as it currently stands. Therefore, we invite you to submit a revised version of the manuscript that addresses the points raised during the review process.  As you will see, the reviewers had several points of clarification that could be addressed in a revised text.

Please submit your revised manuscript within 60 days Jan 30 2025 11:59PM. If you will need more time than this to complete your revisions, please reply to this message or contact the journal office at plosgenetics@plos.org. Please include the following items when submitting your revised manuscript:

We look forward to receiving your revised manuscript.

Kind regards,

John Prensner

Academic Editor

PLOS Genetics

Anne O'Donnell-Luria

Section Editor

PLOS Genetics

Aimée Dudley

Editor-in-Chief

PLOS Genetics

Anne Goriely

Editor-in-Chief

PLOS Genetics

**Additional Editor Comments :**

Regarding Reviewer #3 comment #1, we think that this concern can be addressed through editing the text and adding a stronger Limitations paragraph to the manuscript.

**Journal Requirements:**

3) Please add a full list of legends for your Supporting Information files after the references list.

Potential Copyright Issues:

i) Figure 2B. Please confirm whether you drew the images / clip-art within the figure panels by hand. If you did not draw the images, please provide (a) a link to the source of the images or icons and their license / terms of use; or (b) written permission from the copyright holder to publish the images or icons under our CC BY 4.0 license. Alternatively, you may replace the images with open source alternatives. See these open source resources you may use to replace images / clip-art:

ii) The following Figure contains a logo or branding: Supplementary Figure 3. We are not permitted to publish this under our CC-BY 4.0 license, even with permission. We ask that you please remove or replace it.

5) Thank you for stating that "Parts of Fig. 1 were created using BioRender.com." Please include this in the figure legend.

7) Please ensure that the funders and grant numbers match between the Financial Disclosure field and the Funding Information tab in your submission form. Note that the funders must be provided in the same order in both places as well. Currently, this grant number " (R272-2017–452)" is missing from the Funding Information tab while grant number "R209-2015-3283" provided by the Lundbeck Foundation is missing from the Financial Disclosure box.

Please indicate by return email the full and correct funding information for your study and confirm the order in which funding contributions should appear. Please be sure to indicate whether the funders played any role in the study design, data collection and analysis, decision to publish, or preparation of the manuscript.

**Reviewers' comments:**

Reviewer's Responses to Questions

Reviewer #1: The manuscript "Disentangling the mutational effects on protein stability and interaction of human MLH1" by Larsen-Ledet and Stein reports on a systematic experimental assessment of the impact of MLH1 variants on protein abundance/stability and intraction with its partner PMS2. The study seems well-conducted and the results are presentedin a clear and critical manner. The data themselves are an important contribution to the field and the authors highlight some interesting findings with respect to the evolutionary and functional properties of the protein. I think some aspects of the computational analysis would benefit from a deeper look (see below).

- The authors found that substitutions of surface-exposed residues in the central α-helix spanning position 605-627 exhibited slightly decreased interaction with PMS2, without affecting the MLH1 abundance. They speculated that the central α-helix spanning position 605-627 could be a yet unknown binding site for PMS2, and tested this hypothesis by running AF on MLH1 and a fragment of PMS2 (supp fig. 10). What is the Predicted Aligned Error (PAE) corresponding to that interface? Is it higher than the interface obtained with the full-length PMS2? More generally, PAE is a better metric than pLDDT to assess the quality of complex interface, and hence it would be desirable to report the PAE matrices for the predicted 3D models. The interface predicted TM-score (iPTM) is also now included in ColabFold, the authors may consider reporting this metric as well. It would also be interesting to report the differences between the unbound protein structures and the bound conformations in the predicted 3D model.

- "Interestingly, at position 540, the substitution to glutamic acid had no effect on the interaction with PMS2, whereas the substitution to aspartic acid displayed complete loss-of-interaction". Could the authors rationalise this observation by analysing the network of interactions at the interface?

- Since the authors have produced experimental data for abundance and hence stability, they could

> assess the accuracy of Rosetta score with respect to this phenotype

> replace Rosetta score by the experimental abundance scores in Fig 4B; this would likely extend the main cloud of points and improve interpretation of the results. Are the variants impacting interaction (which would be strongly red or strongly blue) but not stability (they would be on the left) under selective pressure (at the bottom)? Right now, it's a bit difficult to draw conclusions because there are some points with Rosetta scores close to zero (left) and with very negative GEMME scores (bottom) that are red on panel A (indicating strong impact on stability).

- What are the residues with very negative GEMME scores that do not show up in any of the experimental assays. What are their positions in the protein sequence? Do they cluster in 3D so as to form some kind of interface?

Reviewer #2: This manuscript by Larsen-Ledet and Stein investigates the effects of missense mutations on the stability and protein-protein interaction (PPI) of the human MLH1 protein, which is implicated in Lynch syndrome. The authors use two high-throughput yeast-based assays to systematically examine the effects of 4839 (94%) MLH1 missense and nonsense variants in the C-terminal domain on protein abundance (DHFR-PCA) and interaction (Y2H) with its obligate binding partner, PMS2. One of the study’s key finding is that the majority of MLH1 variants that lose interaction with PMS2 also reduce MLH1 stability, suggesting that the loss of interaction might be difficult to separate from loss of stability and reduced cellular abundance. This suggests a strong link between protein stability and interaction for MLH1 and the authors propose that this could be a generally feature of interactions that are driven by large and well structured regions. The authors propose a novel distal region in the C-terminal domain of MLH1 where substitutions can cause both decreased and increased binding with PMS2, independent of changes in abundance. This region is proposed to be a novel interaction site. The authors also compare their experimental data with computational predictions of thermodynamic stability (using Rosetta) and evolutionary conservation (using GEMME) and find a good qualitative agreement between the predictions and experiments but do not really quantify these findings. Lastly, the authors analyse the clinical relevance of their findings by comparing the experimental data to variant classifications in the ClinVar database and population allele frequencies from gnomAD. They again qualitatively sow that their experimental data can distinguish between benign and pathogenic MLH1 variants with some curious exceptions.

The most significant contribution of this manuscript is the generation of comprehensive datasets that provide mechanistic insights into the impact of missense mutations on MLH1. This provides a valuable resource for understanding MLH1 function and dysfunction. The impact of interface mutations on protein stability also provide an observation that is worth noting as a potential more general trend for future studies with a larger set of protein interactions. This work is generally well-executed and primarily lacks some additional quantifications for some of the observations made in many sections of the work.

Major Concerns:

1 - Quantification of Relationships: The manuscript makes many qualitative statements or observations that require quantitative support. None of these quantifications are themselves a major concern but overall the manuscript is missing too many of these. Below is a list of examples that I think would benefit from quantification and statistics:

1.1 - AlphaFold2-Multimer iPTM score for the predicted structure and the similarity with the yeast experimental structured.

1.2 – The iPTM score for AlphaFold2-Multimer of the PMS2 α-helix interacting with the central α-helix in MLH1

1.3 – line 202 - “median abundance scores correlated with relative solvent accessible surface area (rSASA)” add correlation coefficient and p-value

1.4 – line 224 “we performed low-throughput Y2H growth assays” – please describe the results of these experiments in the main text

1.5 – lines 268-275 :”We mapped the experimental abundance scores to the predictions, which revealed that variants with wild-type-like abundance clustered in the upper left corner, whereas low-abundant MLH1 variants mapped towards the bottom right corner”. Please quantify this – what is the correlation between GEMME, predicted ddG and abundance change, similar for the interaction change in the continuation of the text. One might expect that ddG is better predictor of abundance tan GEMME. Is tis true ?

2 - Predictive Power for Pathogenic Variation: While the manuscript show that the data can qualitatively distinguish between benign and pathogenic variants, it lacks a detailed analysis of the predictive power of the abundance and interaction scores for pathogenicity. The authors should calculate performance metrics such as sensitivity, specificity, and area under the receiver operating characteristic curve (AUC) to quantify how well their data can predict pathogenic variants. This would provide a clearer picture of the clinical utility of their findings. This concern relates to the first but it is important to make this part particularly clear.

2.1 - Please quantify the relation between the experimental results and the human genetic (ClinVar variants and allele frequencies). Are the experimental results predictive of allele frequency and pathogenicity ?

2.1 - Are the prediction models (from Rosetta and GEMME) better or worse in terms of predicting allele frequency and pathogenicity ?

2.2 - Can the experimental results (stability and interaction) be combined to improve the prediction of the human genetics data ?

2.3 - Can the experimental results be combined with the prediction models to predict the human genetics data ?

Minor points

line 200: “Multimer-predicted structure of the human C-terminal MutLα complex (Fig. 3B). This revealed that most of the sensitive positions were pointing inwards” – it was not clear to me what “inwards” means here

lines 215-221 – There are several structural observations that are made here and I was intrigued by them but it would have been nice to have some ideas by the authors about what might be the underlying structural mechanisms for these examples.

In regards to the relation between Rosetta ddG and protein abundance it would be interesting to discuss where there might be some discrepancy between predicted loss of stability and loss of protein abundance. In other words, are there meaningful numbers of mutations that are predicted to be destabilizing but do not result in lower abundance ? Are these mostly potential issues with the Rosetta predictions ?

It is very strange that there are mutations that decrease stability without impact on stability. The most likely scenario is that the Y2H is mostly acting as a qualitative measurement of interaction and a relatively small amount of protein might be enough to give a positive interaction result. This is mentioned in the discussion and it is very important aspect that may limit some of the conclusions of the work. This is framed in the discussion as the Y2H being more sensitive which was not very obvious and clear to me. Perhaps this part needs to be revised a bit more for clarity.

Reviewer #3: Larsen-Ledet and Stein used a yeast screening system to address two molecular effects of practically all possible missense substitutions of the MLH1-C-terminal domain. This domain confers the constitutive dimerization with its heterodimeric partner PMS2. They assessed the consequences of variants using two biochemical readouts: protein stability (DHFR-PCA) and dimerization (yeast two-hybrid).

They were able to confirm certain aspects that may be expected based on traditional knowledge in protein biochemistry: for example, that more detrimental effects of substitutions are conferred in internal residues and in residues located in secondary structures (e.g. PMID 33106425).

The strength of the work is the large number of variants analyzed in this screening which is of interest for those searching for ways to unveil the functional roles of specific residues or the potential of a given human variant to inactivate the protein and thereby be causative for Lynch syndrome. This may facilitate Lynch syndrome diagnosis of uncertain variants (VUS). A general caveat in the interpretation, however, is that “only” two aspects of potential dysfunctionality/pathogenicity (stability and dimerization) have been addressed (and not catalytic MMR activity, for example) and therefore variant neutrality cannot be assumed solely based on negative findings in the assays performed.

Some issues, therefore, need be addressed to render the manuscript complete and accurate for publication.

Major comments

Major comment #1

Abstract and throughout the manuscript.

The claim that a yeast-based functional assay provides clinically applicable decisions (“benign” and “pathogenic”) goes too far considering that not all possible mechanisms of pathogenicity are being investigated, and that a non-human experimental system is being used (see also https://www.insight-group.org/criteria/ for the procedures used to turn experimental data in clinically applicable decisions). Moreover, the MMR activity, which provides the most complete read-out of the biochemical functionality of a variant MLH1 protein, is not included in this work, neither the endonuclease activity, which is encoded in the MLH1-PMS2 CTD (this activity should also be reported in line 255 of the manuscript and not omitted).

Therefore, the claim to measure whether a variant is neutral or pathogenic must be dropped, it can only be stated if the variant is “functional” or “non-functional” in terms of stability and dimerization.

Major comment #2

Lines 132ff. Sensitivity and dynamic range were confirmed with variants supposedly neutral. However, for Q542L, which is a VUS, conflicting results have been published, including those that suggest an MMR defect (e.g. PMID 20533529, discussed in PMID 36054288), an information that should be included in the discussion.

Moreover, the authors state correctly that variants in the N-terminal domain cannot be expected to directly impact interaction (lines 154 ff.) However, they detected a complete loss of interaction for the T117M variant located in the N-terminal domain (Fig. 2A, lines 146 ff.). Therefore, it must be concluded that Y2H assay detects loss of interaction as well as loss of stability. Consequently, a variant that is deficient in both assays is not really a bona fide dimerization-deficient variant, only those which are normal in DHFR and deficient in Y2H. This should be clearly stated and discussed appropriately.

Major comment #3

Considering the large number of variants tested, it is hardly possible discuss the results with all previous data, but for such work that has explicitly concentrated on human MLH1 stability (PMID 23403630 and 31697235) and dimerization with PMS2 (PMID 20533529) should still be used for comparing the results of the screening to previous measurements, both as a means of verification of the methods and summarizing the current knowledge. Therefore, a some kind of comparison in form of a table or otherwise would add a lot of worthwhile information to the work.

Minor comments

Line 319 ff.

It may be of interest to add the information that SLiMs have also been identified in MLH1, therefore they too constitute a functionality relevant feature in MLH1-PMS2 biochemistry (PMID 36215471, 32731489 and 37224528).

**Have all data underlying the figures and results presented in the manuscript been provided?**

Reviewer #1: Yes

Reviewer #2: Yes

Reviewer #3: Yes

PLOS authors have the option to publish the peer review history of their article (what does this mean? ). If published, this will include your full peer review and any attached files.

**Do you want your identity to be public for this peer review?** For information about this choice, including consent withdrawal, please see our Privacy Policy .

Reviewer #1: No

Reviewer #2: No

Reviewer #3: No

**Figure resubmission:**
---

## [Decision Letter · Decision Letter 1]

24 Mar 2025

PGENETICS-D-24-01287R1

Disentangling the mutational effects on protein stability and interaction of human MLH1

PLOS Genetics

Dear Dr. Stein,

Thank you for submitting your manuscript to PLOS Genetics. After careful consideration, we feel that it has merit but does not fully meet PLOS Genetics's publication criteria as it currently stands. Therefore, we invite you to submit a revised version of the manuscript that addresses the points raised during the review process.

Please submit your revised manuscript within 30 days Apr 23 2025 11:59PM. If you will need more time than this to complete your revisions, please reply to this message or contact the journal office at plosgenetics@plos.org. Please include the following items when submitting your revised manuscript:

We look forward to receiving your revised manuscript.

Kind regards,

John Prensner

Academic Editor

PLOS Genetics

Anne O'Donnell-Luria

Section Editor

PLOS Genetics

Aimée Dudley

Editor-in-Chief

PLOS Genetics

Anne Goriely

Editor-in-Chief

PLOS Genetics

**Additional Editor Comments:**

The reviewers felt that the revised manuscript was much improved but note a few sentences that should be edited for accuracy and clarity. Please address the final reviewer comments and submit a revised manuscript.

**Journal Requirements:**

1) Please upload all main figures as separate Figure files in .tif or .eps format. For more information about how to convert and format your figure files please see our guidelines: 

**Reviewers' comments:**

Reviewer's Responses to Questions

Reviewer #1: The authors have addressed my concerns. Please put the unit (Angstrom) for the RMSD reported on p.6.

Reviewer #2: The authors have address all major concerns that I had previously.

Reviewer #3: Overall, the authors have adequately incorporated the suggested improvements.

Concerning the previous “major point 1”, though the authors have correctly addressed this issue, however, the sentence in the abstract “Our data successfully DISTINGUISH benign from pathogenic MLH1 variants…”, referring to Figure 4, still is rather misunderstandingly suggesting that clinical decisions are a capacity of the applied assays. Please change to make clear that the acquired data CORRELATED with clinical classifications of the investigated variants (as shown in Figure 4), to make sure that this point is clear.

Minor comment:

Figure 4: In the Figure legend, it is stated that variants without gnomAD occurrence are “indicated to the left of the dashed lines”. There is no dashed line, rather the “axis break” in the x-axis is meant here. Please either introduce a dashed line or correct in the legend.

**Have all data underlying the figures and results presented in the manuscript been provided?**

Reviewer #1: Yes

Reviewer #2: None

Reviewer #3: Yes

PLOS authors have the option to publish the peer review history of their article (what does this mean? ). If published, this will include your full peer review and any attached files.

**Do you want your identity to be public for this peer review?** For information about this choice, including consent withdrawal, please see our Privacy Policy .

Reviewer #1: No

Reviewer #2: No

Reviewer #3: No

**Figure resubmission:**
---

## [Editor Report · Decision Letter 2]

8 Apr 2025

Dear Dr Stein,

We are pleased to inform you that your manuscript entitled "Disentangling the mutational effects on protein stability and interaction of human MLH1" has been editorially accepted for publication in PLOS Genetics. Congratulations!

Yours sincerely,

John Prensner

Academic Editor

PLOS Genetics

Anne O'Donnell-Luria

Section Editor

PLOS Genetics

Aimée Dudley

Editor-in-Chief

PLOS Genetics

Anne Goriely

Editor-in-Chief

PLOS Genetics

Comments from the reviewers (if applicable):

**Data Deposition**

http://datadryad.org/submit?journalID=pgenetics&manu=PGENETICS-D-24-01287R2

**Press Queries**

---

## [Editor Report · Acceptance letter]

PGENETICS-D-24-01287R2

Disentangling the mutational effects on protein stability and interaction of human MLH1

Dear Dr Stein,

We are pleased to inform you that your manuscript entitled "Disentangling the mutational effects on protein stability and interaction of human MLH1" has been formally accepted for publication in PLOS Genetics! Your manuscript is now with our production department and you will be notified of the publication date in due course.

With kind regards,

Zsofia Freund

PLOS Genetics

On behalf of:
